# Content analysis of Persian/Farsi Tweets during COVID-19 pandemic in Iran using NLP

**Pedram Hosseini**[1,*]**, Poorya Hosseini**[2]**, and David A. Broniatowski**[1,*]
[1]The George Washington University, Washington, D.C., USA
[2]Flagship Pioneering, Cambridge, MA
*{phosseini,broniatowski}@gwu.edu

## Abstract

Iran, along with China, South Korea, and Italy was among the countries that were hit hard in the first wave of the COVID-19 spread. Twitter is one of the widely-used online platforms by Iranians (Statcounter, 2020) inside and abroad for sharing their opinion, thoughts, and feelings about a wide range of issues. In this study, using more than 530,000 original tweets in Persian/Farsi on COVID-19, we analyzed the topics discussed among users, who are mainly Iranians, to gauge and track the response to the pandemic and how it evolved over time. We applied a combination of manual annotation of a random sample of tweets and topic modeling tools to classify the contents and frequency of each category of topics. We identified the top 25 topics among which living experience under home quarantine emerged as a major talking point. We additionally categorized the broader content of tweets that shows satire, followed by news, is the dominant tweet type among Iranian users. While this framework and methodology can be used to track public response to ongoing developments related to COVID-19, a generalization of this framework can become a useful framework to gauge Iranian public reaction to ongoing policy measures or events locally and internationally.

## 1 Introduction

As COVID-19 has rapidly and widely spread in the United States and globally, this now pandemic is shaking up all aspects of daily life in all countries affected in an unforeseen manner. Economic activities have been disrupted globally on an unprecedented scale (Nicola et al., 2020) and governments are resorting to a number of policies and measures trying to manage primary and subsequent health, economic and financial aspects of this crisis. While each country will have its unique experience of dealing with this pandemic, there are shared aspects in terms of how different societies deal with and react to the spread of the virus. These commonalities stem from the nature of the virus itself and the biological and psychological similarities of humankind regardless of geographical boundaries. Additionally, there is the commonality in terms of policies that have been devised for mitigation and control of the virus across borders. Iran, along with China, South Korea, and Italy has been among the countries that have been hit hard in the first wave of the viral spread the cause of which is to be yet fully explained. Iranians have been using social media outlets such as Telegram, WhatsApp, Twitter, Instagram, and Facebook to both receive a large portion of their daily news, in addition to spreading the information to one another and expressing their opinion about various developments in the country such as social unrest or in this case events and issues related to the COVID-19 spread. Leveraging Machine Learning and Natural Language Processing (NLP) techniques we are conducting an ongoing analysis of the reaction of the Persian/Farsi speaking users[1] on social media starting with Twitter. In this work, we applied topic modeling to find the themes of tweets posted in Persian/Farsi about COVID-19, followed by manual annotation of a random subset of tweets to assess the distribution of various content types among tweets. We believe our framework can be valuable in monitoring public reaction to ongoing developments locally and internationally related to COVID-19 pandemic, but additionally, a tool and platform to be used for future major economical, political, or health-related events among Persian/Farsi users.

---

[1]It is worth pointing out that even though Farsi/Persian is not spoken only by Iranians and there are users from other countries including Afghanistan that may also tweet in Persian, looking at the type of language and topics being discussed, it is fair to assume that the strong majority of Persian speaking users are Iranians.

This study is organized as the following. In Section 2, we briefly outline the data collection and preprocessing steps on tweets. In section 3, we outline the methods and experiments that were applied to raw data to obtain the results and insights. Lastly, we report the insights and results of our data collection efforts and analysis in Section 4. We conclude the paper in Section 5 by summarizing the results and discussing future directions and next steps.

## 2   Data collection

We used the Social Feed Manager (SFM) (Libraries, 2016) platform to collect tweets[2]. SFM is a software developed at the George Washington University that uses Twitter Developer API to help researchers with collecting tweets. We listed a group of hashtags that are associated with COVID-19 related Persian tweets on Twitter. Hashtags and their corresponding English translation are shown in Figure 1. We chose these hashtags based on the trends

| Farsi/Persian hashtag | English translation |
|---|---|
| #کرونا | Corona |
| #کروناویروس | Coronavirus |
| #ویروس_کرونا | Corona_virus |
| #کارزار_کرونا | Corona_campaign |
| #در_خانه_بمانیم | Let's_stay_home |
| #قرنطینه_خانگی | Home_quarantine |
| #کرونا_از_آمریکا | Corona_from_America |

Figure 1: List of Farsi/Persian hashtags related to COVID-19.

on Twitter since we started the data collection. We also added new hashtags associated with Persian COVID-19 on Twitter which we will use in our future analysis. We started tweets collection process since March 12th, 2020 and we're still collecting tweets in real-time so that we can do future processing on tweets and update our results. The results in our experiment section are using tweets from March 13th to the end of April 19, 2020.

### 2.1   Preprocessing tweets

We did some preprocessing on text of tweets before using them for our topic analysis. We only considered the *original* tweets themselves. The reason for choosing original tweets alone is that in many cases, replies and quotes may not either have

text associated with the person who replies or the responses are shorter in length or not as informative as they should be. We also used the *lang=fa* attribute to filter tweets' language and have only those written in Persian. Then, we removed URLs, emojis, and punctuation marks, and English numbers. We also removed any mentions of user screen names in tweets. In the end, we normalized the tweets' text using the normalizer method in Hazm[3] library.

Also, we created a list of Persian stop words specifically for the analysis of this corona related collection of tweets. Even though there are some list of Persian stop words currently available, we created a new list because the definition of a stop word can be different for different tasks across various domains. For example, words such as *hard* or *people* are listed as stop words for topic modeling on some of the available Persian stop word lists while these words can potentially help us to understand the theme of a topic among corona related tweets. The preprocessing step resulted in having **530,249** unique tweets and **43,566** unique tokens. These tweets are the input to our analysis.

## 3   Experiments[4] and methods

In this round of analysis, we use topic modeling, Latent Dirichlet Allocation (LDA), to analyze the topics of original tweets. The main goal here for using LDA is to find the topics that are being discussed in Persian tweets. We discuss this analysis in section 3.1. We also annotated a random sample of tweets from two days to find out what type of content tweets are mostly about. We discuss the details of annotation in section 3.2.

### 3.1   LDA analysis

We used Mallet for LDA analysis (McCallum, 2002) and the Gensim python wrapper methods (Řehůřek and Sojka, 2010). Using the bag of words (BoW), we first built a dictionary and corpus using all the cleaned tweets. Then we set *k=50*, where *k* is the number of topics, and generated an LDA model using the dictionary and the corpus. We also enabled Mallet's hyper-parameter optimization by setting *optimize_interval=10*.

---

[2]We have publicly shared the tweet IDs on our GitHub repository.

[3]https://github.com/sobhe/hazm

[4]Code, reproducibility reports, and jupyter notebooks are documented and publicly available at: https://github.com/phosseini/COVID19-fa

### 3.1.1 Choosing number of topics

For choosing the final value of k, number of topics, we first created multiple LDA models with different values of *k* including {*50, 100*}. Then we manually checked the output of these LDA models. Specifically, we looked at the top words in each generated topic to get an understating of the theme of the topic and compare overlap and similarity of topics. To be on the safe side and not lose any useful information from the generated topics, we decided to set k big enough so that it will cover all the possible topics. The downside of choosing a larger value of k is ending up with more overlapped topics –topics that are similar to one another to a high degree. However, the benefit of choosing a large k is that we will not miss any topic among the tweets. In the end, between k=50 and k=100 we chose 50 since topics in k=100 model were too specific, not informative, and in some cases almost completely overlapped. With k=50, we still have some partially overlapped topics but far less than k=100.

### 3.1.2 Finding dominant topics

In an LDA model, each document[5] is a distribution of topics. We first find the dominant topic in a document. By a dominant topic, we mean the topic with the largest association value with the document. Then, we group documents over the entire corpus by their dominant topic. In this way, we find the topics that are dominant among the majority of documents in the corpus and we call these topics the top topics. The top-25 topics are shown in Figure 2.

### 3.2 Content Analysis

In addition to the specific topics discussed in the LDA analysis, we were interested in broader category of content of the COVID-related tweets. This required a step that involved manual labeling of a representative sample of tweets by two individuals. We started by manually looking at tweets prior to the analysis and topic modeling, and observed that certain categories such as satire and complaint are fairly dominant. Our initial observation and hypothesis were that users, representing Iranians, mainly either blame different entities (e.g. government or fellow citizens), complain about the situation or make jokes more frequently compared to discussing measures for fighting against

---

[5]We use document and tweet interchangeably.

the COVID-19.

For our content analysis, we first used the *Mini-BatchKmeans* algorithm to cluster the tweets from March 12, 2020 to March 14, 2020, total **45,234**, into multiple categories. Using Elbow method, we found *k=8* to be the optimal number of clusters for our analysis. Then we fitted the K-means algorithm to the number of the optimal clusters on the TF-IDF vector of tweets. This process resulted in having one label from range of clusters, {*0,...,7*}, for each tweet.

### 3.2.1 Manual annotation

We randomly sampled 30 tweets from each cluster. We defined a set of categories including: {*"opinion", "news/quotes", "satire/jokes", "complaint/blame", "solution", "neutral"*} and two annotators manually assigned a label from this set to each tweet. These categories are defined/chosen based on the themes we could see among tweets by manually reading and pre-annotating a sample of tweets. We tried to define the categories in a way that they can cover a variety of content types. It is important to mention that *solution* category is related to tweets that are constructive and talk about different ways of fighting the spread of the COVID-19, raising awareness, or giving hope to other users. *Neutral* tweets are mainly tweets that did not belong to any of the other categories or could not be easily understood (e.g. using Farsi to type in local languages.)

To estimate the overall representation of each category of tweets among all the clusters, we multiplied the ratio of each label in each cluster by cluster's weight/ratio. Cluster ratio is the number of tweets in the cluster over the number of all tweets.

### 3.2.2 Inter-annotator agreement

We used *Cohen's kappa* metric from scikit-learn package (Pedregosa et al., 2011) to compute the inter-annotator agreement between annotators. The agreement between annotators is *0.47* which is a moderate and more than fair agreement (Viera et al., 2005). By taking a closer look at samples, we noticed one main reason for disagreement is the challenging nature of the task and many borderline cases. Figure 3 shows some of the disagreement examples. To resolve the disagreement cases, the two annotators discussed those cases together by reading the tweets and looking at the labels that were already assigned but without knowing what label is assigned by who. In many cases, the final

| Topic ID | Percent (%) | Top words | English translation |
|---|---|---|---|
| 4* | 5.54 | کرونا، تموم، بگیرم، قرنطینه، دلم | corona, over, [get infected], quarantine, [I wish] |
| 27 | 4.94 | آمار، مبتلایان، ایران، ویروس، مرگ | statistics, infected, Iran, virus, death |
| 32* | 4.48 | کرونا، کنیم، تموم، کاش، قرنطینه | corona, [let's do], over/end, [I wish], quarantine |
| 20* | 4.46 | کرونا، گرفتن، مردم، هنوز، کم | corona, [got infected], people, still, low |
| 1* | 4.44 | کرونا، خونه، بیرون، قرنطینه، کار | corona, home, outside, quarantine, work |
| 28* | 3.87 | کرونا، زنگ، دوست، پدر، خوب | corona, phone call, friend, dad, good |
| 42* | 3.22 | کرونا، فکر، دنیا، کار، بیشتر | corona, think/thought, world, work, more |
| 25 | 3.16 | کرونا، ویروس، شیوع، ایران، چین | corona, virus, spread, Iran, China |
| 2 | 2.72 | کرونا، ایران، آمریکا، ترامپ، مردم | corona, Iran, United States, Trump, people |
| 24* | 2.60 | کرونا، ویروس، درباره، اخبار، خبر | corona, virus, about, news |
| 0 | 2.58 | کرونا، ستاد، تهران، رئیس، مجلس | corona, committee, Tehran, chairman, parliament |
| 37 | 2.37 | کرونا، دنیا، جهان، ویروس، تمام | corona, world, virus, over/end |
| 6* | 2.36 | کرونا، مردم، قرنطینه، کار، دولت | corona, people, quarantine, work, government |
| 39 | 2.33 | کرونا، ویروس، افغانستان، مثبت، هرات | corona, virus, Afghanistan, positive, Herat |
| 7 | 2.30 | قرنطینه، کرونا، خانگی، عید، امسال | quarantine, corona, house, [New Year], [this year] |
| 11 | 2.06 | کرونا، شیوع، ویروس، قرنطینه، اعلام | corona, spread, virus, quarantine, announcement |
| 43* | 1.95 | کرونا، دوستان، لطفا، فیلم، دعا | corona, friends, please, movie, prayers |
| 10 | 1.95 | خانه، کرونا، بمانیم، شکست، جدی | home, corona, stay, defeat/beat, serious |
| 17* | 1.85 | کرونا، خدا، دست، دین، علم | corona, god, hand, religion, science |
| 49 | 1.84 | کرونا، میلیون، کمک، دلار، میلیارد | corona, million, help/aid, dollar, billion |
| 46 | 1.83 | کرونا، پزشکی، درمان، کادر، مبتلا | corona, medical, treatment, staff, infected |
| 13 | 1.81 | کرونا، سلامت، مبارزه، ایران، مردم | corona, health, fight, Iran, people |
| 47 | 1.77 | کرونا، بهداشت، وزارت، سازمان، جهانی | corona, health, ministry, organization, global |
| 45 | 1.76 | کرونا، ایران، ویروس، خامنه ای، اسلامی | corona, Iran, virus, Khamenei, Islamic |
| 30 | 1.73 | کرونا، تشخیص، کیت، داروی، ویروس | corona, diagnose, kit, medicine, virus |

Figure 2: **Top-25** distributed topics over the entire corpus of tweets in descending order. *Topic ID* is the identifier of the topic in LDA model. *Top words* are the top associated words which each topic. *Percent* is the ratio of the number of tweets in which the topic is dominant to the number of all tweets. Starred (*) Topic IDs are those which have the most number of non-zero associations with words in the LDA dictionary.

label was chosen from one of the two different assigned labels. Also, in few cases, annotators agreed that the label for a tweet is different than the labels that were already assigned.

| Example | Annotator 1 | Annotator 2 |
|---|---|---|
| پارسال هی گفتین سبزه نزارید سبزه نزارید بفرما امسال جاش کرونا سبز کردید
Last year everybody said don't grow Norouz Sprouts, don't grow Norouz Sprouts, now what, this year you grew Corona instead | Satire | Complaint |
| اگه کرونا ما رو نکشه قطعا استرس گرفتش نابودمون میکنه
Even if the Corona doesn't kill us, the stress of getting infected will kill us instead | Satire | Opinion |
| هزینه تست کرونا در آمریکا هزار و سیصد دلار علی برکت الله برسونید به گوش براندازها
Corona Testing in the USA costs $1300, let it be noted by US-loving regime opposition | Opinion | Complaint |

Figure 3: Example of some borderline/challenging Farsi tweets on which there was a disagreement between annotators.

## 4 Discussion and Results

In this round of analysis, we report our findings and insights from analyzing tweets on three areas. We first share our insights about level of activity of the users who tweeted in Persian around COVID-19 over time. Next, we look at the correlation between different types of COVID-related tweets with the official number of confirmed cases in Iran. Lastly, we outline the result of our analysis on the content of the tweets to show what topics were mostly talked about among the users around COVID-19 during this ongoing crisis. Related to the topic and content, we additionally broke down the tweets in terms of the type of the language users used when responding to COVID-19.

### 4.1 Tweet activity and confirmed cases over time

As the first step in our analysis, we looked at the volume of the tweets on COVID-19 pandemic over time. To achieve this, we extracted the tweets in Persian that had hashtags related to COVID-19 [de-

tailed in section 2]. Figure 4 shows the number of all COVID-19 related tweets in the span of nearly five weeks since the onset of the crisis in Iran. By looking at Figure 4, we see a dramatic decrease in the number of tweets as we get closer to March 20, the Nowruz or Persian New Year, which is also the first day of Spring. This could indicate an increase in travel and trips around the Persian New Year as users tend to be less active on Twitter around this time. At this point, there was no official enforceable policy in place to limit such road trips and travels across the state lines.

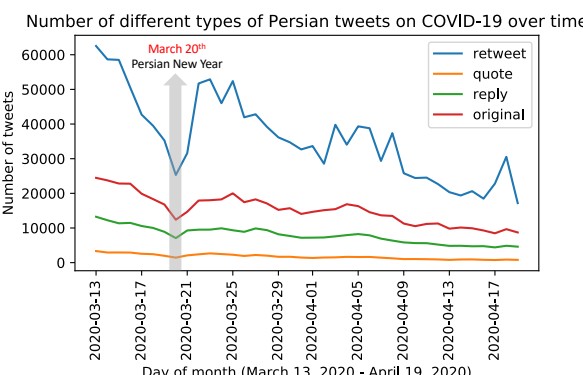

Figure 4: Number of original, retweet, reply, and quotes Persian/Farsi COVID-19 related tweets in March and early April. March 20$^{th}$ was Nowruz, the Persian New Year, the first day of Spring.

We additionally looked into the correlation between the volume of the COVID-related tweets and number of confirmed COVID-19 cases in Iran. To do that, we extracted the number of confirmed, death, and recovered cases in Iran from the official website of the Ministry of Health and Medical Education[6]. Figure 5 shows the number of confirmed, death, and recovered cases in Iran. When we look at the number of posted tweets and confirmed cases during the same period of time in Figures 4 and 5, respectively, we notice that even though the situation was not getting better in terms of the number of confirmed COVID-19 cases in Iran and country had not been reached to the peak of the pandemic, conversation and tweets on COVID-19 had already

---

[6]http://corona.behdasht.gov.ir/ access may be restricted for users outside Iran. Some dates and numbers on the website were missing during this time window where we instead used numbers reported in Johns Hopkins CSSE Data Repository at https://github.com/CSSEGISandData/COVID-19. In cases where there was a conflict between these two sources, we used the numbers reported on the ministry's official website. It is worth mentioning that these conflicts do not have any major impact on the overall trend and thus conclusions of this paper.

been started to decrease. It is worth to further study the reasons behind such phenomenon. For example, is less conversation on COVID-19 due to simply losing interest in the topic or is it because of the underestimation of the pandemic by Iranians and lack of understating of the concept of flattening the curve. We should further add that the number of confirmed cases may be different than the actual number of infected people any time point due to inefficacy in testing procedures especially during early phases on the infection.

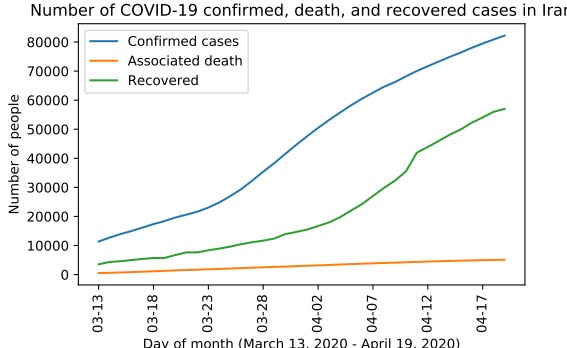

Figure 5: Official number of Corona-virus (COVID-19) confirmed, death, and recovered cases.

## 4.2 Popular topics among the tweets

We conducted Latent Dirichlet Allocation (LDA) (Blei et al., 2003) topic analysis on the collection of tweets to identify major COVID-19-related topics among Persian-speaking users. In Figure 2, we have listed the top-25 topics as well as listing the top words associated with each of the topics. By looking at the top words and strongly associated tweets with each topic in our LDA models, we analyzed most popular topics among the tweets further among which life experience of living under home quarantine was the dominant one. In the following paragraphs, we summarize our findings and share our insights on a few selected topics.

### 4.2.1 Experience of living under home quarantine is the dominant topic

As alluded to earlier, life experiences a result of living under quarantine is a major discussion topic. Users mainly talk about what they wish they could do but now cannot because of their new life style. There is a clear sign of frustration and fatigue as well as complaining about the life under quarantine. There are users who blame their fellow citizens for not taking the situation seriously. For

instance, some users blame those who did not follow the quarantine and celebrated Chaharshanbe Suri –Iranian festival of fire celebrated on the eve of the last Wednesday before Nowruz. Another significant theme among such tweets is the feeling of helplessness and despair. There are additionally tweets on how individuals miss visiting their parents, siblings, or relatives. The feeling of depression is not specific to the users in this study and is a mutual problem in many countries that are facing the crisis (Rajkumar, 2020), however, it is not still clear if such feelings are being taken seriously or if there is any plan to help people who face depression with psychotherapy or counseling. Last major theme is a significant number of satirical tweets that in many cases are combined with blames and complaints.

Another major theme is not surprisingly tweeting about the news and reports. The topic of many such tweets are the latest on the number of confirmed infected, death, and recovered cases both in Iran and abroad with Italy, Spain, and United States being among the top countries. Interestingly, there is a discussion around lifting of the US sanctions against Iran. Tweets associated with this topic are coming from users who are both pro or against lifting the sanctions. Some argue that US should lift the sanctions to help Iranian people overcome the COVID-19 crisis while a majority of tweets are about why US should not lift the sanctions. Another major topic are tweets about Afghanistan, a country that is experiencing more number of COVID-19 related cases recently where pandemic is affecting the country with some delay after Iran.

Lastly there are also both pro- and anti-Iranian regime tweets (*Topic 45*). For the pro-regime, for example, users praise Basij, a paramilitary forces of the Islamic Revolutionary Guard Corps, who took action to mitigate the crisis or quote the supreme leader of Iran who praises Iranian people's effort and cooperation in fighting against COVID-19. On the anti-regime side, some users blame Ayatollahs and the regime for taking actions that helped spread of the virus and for not taking adequate measures for handling the crisis following the spread. We additionally see tweets from the anti-regime opposition groups who see this crisis as an opportunity to overthrow the Iranian regime, a goal they have been pursuing over recent years and prior to COVID-19 pandemic.

## 4.3 Tweet categories: Satire is dominant followed by news

Figure 6 shows the distribution of different content type over the entire tweets. Results to a certain degree are validating our hypothesis, the fact that users make jokes about the situation rather than talking about how they can fight against the virus. Even though we certainly need to annotate and analyze more tweets for better generalization of these results, still it is important to think about the reasons behind such a phenomenon. It would be interesting to know why satire is the top content category. Here are a two possible explanations, (1) An underestimation of the scale and seriousness of COVID-19 pandemic that was also reflected in the tone of the officials in early stages of the crisis, (2) Nuances of the Iranian culture in using satire as a way of coping with unpleasant realities.

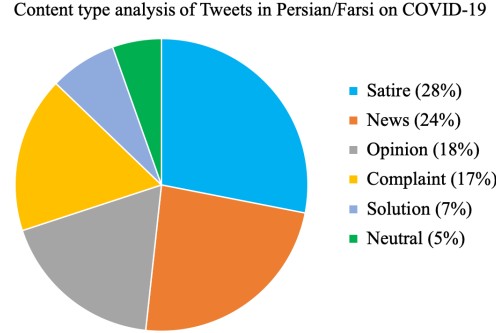

Figure 6: Content analysis of more than 45,000 tweets in Persian on COVID-19

## 5 Conclusion and next steps

In this paper, we collected more than 530,000 original tweets in Persian/Farsi related to COVID-19 pandemic over time and analyzed the content in terms of major topics of discussion and the broader category of tweets. We applied a combination of manual annotation of a random sample of tweets and topic modeling tools to classify the contents and frequency of each category of topics. We identified the top 25 topics among which living experience under home quarantine emerged as a major talking point. We additionally categorized broader contents of tweets that show satire, followed by news, is the dominant tweet type among the Iranian users.

There are a few next steps and directions that we are actively working on. 1) We are interested to understand how COVID-19 related conversations

among Iranian users shifted and evolved over time by tracking discussion topics within specific time windows, 2) Continue to conduct manual annotation of tweets for our topic analysis, as a continuation of content analysis described in section 3.2, for the newly added data points we collect. We expect these annotations to help us to better understand and measure public's reaction to the pandemic and to the specific events that unfolded over time, 3) we are interested to conduct a deeper analysis on the factuality of news and information shared on twitter and categorizing the different types of information in terms of validity and accuracy. For instance, we have observed examples of wrong information and claims about vaccines or treatments for corona-virus in the early days creating a black market for sales of these products. We are running more analysis to initially find the themes of such false information and additionally identify different strategies that are being used for spreading it.

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
