# OpenReview forum: "Content analysis of Persian/Farsi Tweets during COVID-19 pandemic in Iran using NLP"
_EMNLP/2020/Workshop/NLP-COVID — NLP-COVID19-EMNLP Poster_

### Official Review · AnonReviewer1 · 2020-09-15
**Interesting nuggets, recommend to simplify**

**Rating:** 6
**Confidence:** 4

**Review:**

This is a report on the conclusions from a data analysis performed on 530k tweets over a period of one month in Persian language.

[strengths]
 - as the authors said, it can indeed become a blueprint for how to perform similar analysis in different countries
 - it contains several interesting nuggets which are probably distinct of this culture, and are worth comparing with studies performed in other languages (eg: the drop during Persian new year, the high tweets which are about satire)

[weakness]
 - there are several points which are unclear (see below), and I believe should be clarified if this is to be accepted
 - it contains several unrelated analyses and reads sometimes as a collage of different things. I would encourage the authors to trim the paper down (eg: focus either on the LDA or kmeans, remove Appendix B)
 - the paper would be easier if read in reverse: data-collection comes at the end and lots of material is repeated (first the conclusion (Sect 2) and then again when explained the process (Sect 3) ). While this might be standard in other disciplines, it is not the case in NLP. I think that by working on the structure the paper could be made clearer and to the point.


[clarification questions/comments]
 - is Fig 2 just the zoom of the red curve of Fig 1? If so, it could be removed
 - "Twitter is one of the widely-used online platforms by Iranians" could you add a reference to that, detailing, for instance, the market penetration?
 - Sect 3.1.2: for what is the document-wide analysis done? If you want a hard-clustering, then why do an LDA and not directly a kmeans like you do afterwards? In general, the LDA analysis seems underexploited. Was it used to obtain the categories of 3.2.1?
 - you detail the features for LDA , but not for kmeans
 - I think your extrapolation of classes from a random sample of 30 of each category has issues. Depending on the size of the cluster the error might be high (increased by your IAA). Why not train a classifier with the annotations, maybe adding some weakly-supervision? This would also strengthen the technical aspects of the paper.


As a conclusion, I propose to reduce the paper lengths substantially by focusing on the main contribution (which for me is the annotated dataset and the clustering), remove duplicate description (which could be easily done by reverting the current order: Sect 4 first, then Sect 3 and finally Sect 2 with the plots) and deciding what to do with the LDA analysis (the appendix might be a good place if it was only used to obtain the categories)

---

### Official Review · AnonReviewer3 · 2020-09-18
**The paper need to be restructured, proofread and trimmed.**

**Rating:** 5
**Confidence:** 3

**Review:**

The comments I am making are just some examples of the issues I found with the paper – I believe the authors would benefit from collaborating with writing experts from their university.

The paper needs to be proofread. There are spelling mistakes (i.e. loosing, fairy), grammatical errors (i.e. this study is organized as following), missing punctation and long sentences.

The structure of the paper is in reverse of usual order in this discipline. It should be changed to: Introduction, Data, Method, and Analysis and Conclusion.

There are references to Tables in the paper, but every image is captioned as a Figure.
The fact that Twitter is widely used by Iranians needs a reference. It is hard to believe this platform is popular between Iranians given their aversion to government oversight and their private nature. (What percentage of Iranians use Twitter? Compared to Facebook/Instagram/Pinterest?)

Section 2:
It is strange to read first “Discussion and Results” before any explanations, and phrases such as “In this round of analysis” are perplexing  - at this point we have no notion of “rounds of analysis” let alone what this round might be.
The section talks about tweets related to COVID-19 and tells us to read section 4 to get an insight into the process. If we read section 4 we find that we must refer to Figure 7 to see what search terms were used – and we find that none of the English translations use the term “COVID” – rather they mostly all use variations of “corona”. Do Iranians not use the COVID-19 terminology? This needs an explanation.

Section 2 brings to our attention the fact that the number of corona-related tweets has steadily gone down after the onset of the disease in Iran.  However, there is no analysis done on this important trend, instead an obvious point is made about people being less active on Twitter around the Persian New Year day! This diminished activity is explained as related to travel – is travel the only thing that Iranians do more of that would explain the pattern around Nowruz – but really, why is this significant at all for the study? It could just be quickly mentioned and not in such detail. Figure 2 seems to be redundant as the trend is already in Figure 1 and because of the lack of importance to Nowruz for the study – could the annotation of March 13 be added to Figure 1?

Section 2.3, Figure 5 presents a clear tweet distribution over some topics but the discussion before the figure does not clearly explain where the distribution comes from. The discussion seems to imply that the numbers in Fig 5 are based on manual labelling. The reader is left asking herself how many tweets were annotated if manual labelling was used. Or, if not, then how were these 6 distinct categories derived? We must detour to the suggested detail in the later section 3.2 to understand these are related to a k-means clustering content analysis process. This is just one example of the frustration that occurs because of an incorrect sequence. This is also confusing because this should be explained after the annotation section.

Section 3:
The top 25 topics out of 50 topics are listed and some are analysed here. Looking at the list of topics they seem to be very overlapping. It seems that the number of 50 for topics is too high. On the other hand, references are made that would require a fine distinction between topics, but these cannot be found in the examples. For instance, Basij is mentioned as one of the top topics but it is not obvious which topic in Figure 4 it is related to. Should we have seen more of the top words to understand which of the pro or anti regime topics contains this reference? We do not need to keep seeing the word “corona” in every topic – common words would normally be eliminated during topic modelling but can at least be removed from the table so we can see more of the distinctive words.

Section 3. The choice of 50 topics seems unreasonable and it is not justified. Looking at the top 25 topics, we could merge some.

Section3.2 Both k-means clustering, and topic modelling seem to be used for the same analysis. I would suggest to stick to just k-means clustering and analyse the taxonomy of Persian corona-related tweets.
The k-means clustering is done on 8 categories but annotation is done with 6 labels. This needs explanation. Give the number of annotated tweets. If it is only 240 tweets that is not enough for giving a sound analysis.

Section 4.1 The keywords used for collecting tweets only based on 8 hashtags which is a very limited scope.

Section 5. It would have been useful to show a few examples of satire tweets.
Appendices. The mentions of conversations on Whatsapp is completely irrelevant and confusing for the reader. Whatsapp is a messaging app, not belonging to the social media category.

---

### Official Review · AnonReviewer2 · 2020-09-19
**Interesting reading, but should be professionally edited**

**Rating:** 5
**Confidence:** 3

**Review:**

This paper presents an analysis on tweets posted in Persian/Farsi about COVID-19, including
 -- the correlation between tweet discussions and infected cases
 -- topic analysis using Latent Dirichlet Allocation
 -- tweets categorization into several manually defined categories

Strengths
 -- It is a really interesting reading, providing lots of interesting observations on Iranian public reaction

Limitations
 -- The paper structure looks strange, and it causes difficulty in understanding the paper. Should Section 2 (results) be put after section 3 (methods)? and Section 4 (data collection) become Section 2? Put Appendices after References. Don't feel Appendix B.1 fits the paper.
 -- The writing can be improved, especially when describing methods. Add citation and explain in details methods.
 -- The experimental setup looks arbitrary. For example, the explanation in Section 3.1.1 and 3.2 are lack of detail and not convincing.

Other suggestions
- Section 2.2.1: When giving examples about these topics, it would be better to refer to Table 4, explaining why it is the case
- Table 6, suggest to add English translations
- It seems a paper that should have broder group of readers (ordinary people) rather than only NLP researchers.

---

### Author Response · Authors · 2020-09-27
**Constructive comments, sections are re-ordered + more details provided for experiments**

We want to start by thanking the reviewers for their constructive comments. We believe we can address the majority of them shortly and they will certainly contribute to making the paper significantly better. To cite one example, placing the “results and discussion” section after the analysis section would be probably more appropriate as suggested. We have already updated our paper accordingly. We initially thought having an overview of the results at the beginning of the paper may be more interesting for pre-publication on arXiv.

Additionally, we’d like to highlight that in addition to providing as many details as possible about experiments and methods in the paper itself to make it self-explanatory, we have also documented all experiments and reproducibility reports for every single experiment in jupyter notebooks that are publicly available on our GitHub repository at https://github.com/phosseini/COVID19-fa

Lastly, we’d also want to highlight one of the major contributions of our paper in that it provides one of the largest Twitter collection IDs and COVID-19 related hashtags in Farsi, that are constantly being updated since March 2020, that we believe will be of great importance for researchers who want to continue to analyze social media data around a variety of COVID-19 related topics among Iranians and Farsi/Persian speaking users on Twitter.